# Effects of Ibuprofen Use on Lymphocyte Count and Oxidative Stress in Elite Paralympic Powerlifting

**DOI:** 10.3390/biology10100986

**Published:** 2021-09-30

**Authors:** Felipe J. Aidar, Guacira S. Fraga, Márcio Getirana-Mota, Anderson Carlos Marçal, Jymmys L. Santos, Raphael Fabricio de Souza, Alexandre Reis Pires Ferreira, Eduardo Borba Neves, Aristela de Freitas Zanona, Alexandre Bulhões-Correia, Paulo Francisco de Almeida-Neto, Tulio Luiz Banja Fernandes, Nuno Domingos Garrido, Maria do Socorro Cirilo-Sousa, María Merino-Fernández, Alfonso López Díaz-de-Durana, Eugenia Murawska-Ciałowicz, Breno Guilherme de Araújo Tinoco Cabral, Filipe Manuel Clemente

**Affiliations:** 1Graduate Program of Physical Education, Federal University of Sergipe (UFS), São Cristovão 49100-000, Brazil; guacirafraga@yahoo.com.br (G.S.F.); marcio_getirana@hotmail.com (M.G.-M.); acmarcal@yahoo.com.br (A.C.M.); jymmys.lopes@gmail.com (J.L.S.); raphaelctba20@hotmail.com (R.F.d.S.); 2Group of Studies and Research of Performance, Sport, Health and Paralympic Sports (GEPEPS), Federal University of Sergipe (UFS), São Cristovão 49100-000, Brazil; 3Department of Physical Education, Federal University of Sergipe (UFS), São Cristovão 49100-000, Brazil; 4Graduate Program of Physiological Science, Federal University of Sergipe (UFS), São Cristovão 49100-000, Brazil; 5College of Physical Education and Exercise Science, University of Brasília (UnB), Brasília 70910-900, Brazil; alexandreispf@gmail.com; 6Graduate Program in Biomedical Engineering, Federal Technological University of Paraná (UTFPR), Curitiba 80230-901, Brazil; eduardoneves@utfpr.edu.br; 7Department of Occupational Therapy, Federal University of Sergipe (UFS), Lagarto 49400-000, Brazil; arisz_to@yahoo.com.br; 8Department of Physical Education, Federal University of Rio Grande do Norte, Natal 59064-741, Brazil; alexandrebulhoescorreia@gmail.com (A.B.-C.); paulo220911@hotmail.com (P.F.d.A.-N.); brenotcabral@gmail.com (B.G.d.A.T.C.); 9Institute of Physical Education and Sport, Federal University of Ceará (UFC), Fortaleza 60020-181, Brazil; banja@ufc.br; 10Research Center in Sports Sciences, Health Sciences and Human Development (CIDESD), University of Trás-os-Montes e Alto Douro, 5001-801 Vila Real, Portugal; ngarrido@utad.pt; 11Graduate Program Association of Physical Education, Federal University of Paraíba (UFPB), João Pessoa 58051-900, Brazil; helpcirilo@yahoo.com.br; 12Department of Physical Education, Regional of University (URCA), Crato 63105-010, Brazil; 13Faculty of Health Sciences, Universidad Francisco de Vitoria (UFV), 28223 Madrid, Spain; m.merino.prof@ufv.es; 14Sports Department, Physical Activity and Sports Faculty-INEF, Universidad Politécnica de Madrid, 28040 Madrid, Spain; alfonso.lopez@upm.es; 15Physiology and Biochemistry Department, University School of Physical Education, 51-612 Wroclaw, Poland; eugenia.murawska-cialowicz@awf.wroc.pl; 16Escola Superior Desporto e Lazer, Instituto Politécnico de Viana do Castelo, Rua Escola Industrial e Comercial de Nun’Álvares, 4900-347 Viana do Castelo, Portugal; filipe.clemente5@gmail.com; 17Instituto de Telecomunicações, Delegação da Covilhã, 1049-001 Lisboa, Portugal

**Keywords:** Paralympic Powerlifting, ibuprofen, muscle strength, oxidative stress, recovery of function

## Abstract

**Simple Summary:**

Paralympic Powerlifting (PP) is a strength sport and training tends to promote fatigue. Ten national-level PP athletes were evaluated concerning post-training oxidative stress using Ibuprofen and a placebo. Strength indicators were evaluated. The training consisted of five sets of five repetitions (80–90% 1-Repetition Maximum) in the bench press. The IBU had a positive effect on strength indicators, with decreased fatigue and increased lymphocyte count. There were no differences in oxidative stress. The use of IBU provided improvements in strength and fatigue reduction and did not protect against oxidative stress.

**Abstract:**

Background: Paralympic Powerlifting (PP) training tends to promote fatigue and oxidative stress. Objective: To analyze the effects of ibuprofen use on performance and oxidative stress in post-training PP athletes. Methodology: Ten national level PP athletes (age: 27.13 ± 5.57) were analyzed for oxidative stress in post-training. The study was carried out in three weeks, (1) familiarization and (2 and 3) evaluated the recovery with the use of a placebo (PLA) and ibuprofen (IBU), 800 mg. The Peak Torque (PT), Torque Development Rate (TDR), Fatigue Index (FI), reactive substances to thiobarbituric acid (TBARS) and sulfhydryl groups (SH) were evaluated. The training consisted of five sets of five repetitions (80–90%) 1-Repetition Maximum (1-RM) in the bench press. Results: The IBU showed a higher PT (24 and 48 h, *p* = 0.04, ɳ^2^ *p* = 0.39), a lower FI (24 h, *p* = 0.01, ɳ^2^*p* = 0.74) and an increased lymphocyte count (*p* < 0.001; ɳ^2^*p* = 4.36). There was no change in oxidative stress. Conclusions: The use of IBU provided improvements in strength and did not protect against oxidative stress.

## 1. Introduction

Paralympic Powerlifting is characterized by being a sport that demands high intensities during competitions and training, and the training demands progressive overloads to take athletes to the peak of the required physical performance [1,2]. Due to training overloads, it is necessary to perform an adequate recovery so that athletes have performance gains. When insufficient recovery occurs, tissue injury may be induced and may lead athletes to overtrain (loss of performance due to the accumulation of training shifts without adequate recovery) [1,3].

It is noteworthy that intense physical exercise is a physiological stress capable of altering immune responses and blood biomarkers [4]. Scientific studies show that intense physical exercise can modulate the leukocyte count in the bloodstream and the interaction of these leukocytes (neutrophils and monocytes/macrophages) with endothelial cells in the muscle and consequent transmigration to the damaged skeletal muscle tissue [5,6].

Physical exercise, immune system and oxidative stress indicate that volume and intensity are directly related to alterations in the redox balance, and the excessive increase in production or the reduction of antioxidant capacity, which can induce oxidative damage to lipids, proteins and nucleic acids [7,8]. Excessive loads of physical exercise can generate oxidative stress, considering that physical exercise can promote the formation of ROSs (i.e., reactive oxygen species) in the human body [9,10]. It is noteworthy that ROSs can cause tissue damage and, in high concentrations, damage cellular organelles, nucleic acids, lipids and proteins, causing harm to human health [6]. In the same direction, it has been reported that strenuous exercise, as in strength training, tends to increase stress biomarkers. Thus, oxidative stress has been associated with strength training [11,12,13]. Exercise-induced oxidative stress has been associated with reactive oxygen species (ROSs), especially during exercise [14], as well as post-intensive exercise muscle damage and inflammation that tend to contribute to increased oxidative stress [12,15]. In this sense, it has been suggested that elite powerlifters may benefit from blunted responses to oxidative stress after intensive weightlifting sessions, which may have implications for recovery between training sessions (Ammar et al., 2017a). Therefore, to reduce oxidative stress and protect athletes’ bodies with the objective of enhancing the recovery process, many methods have been proposed and used, among which ibuprofen (IBU), which is a non-steroidal anti-inflammatory drug (NSAID), stands out for self-administrative use [1,16]. However, the use of NSAIDs can inhibit the muscle myofibers regeneration, the proliferation and the differentiation of satellite cells, and muscle hypertrophy induced by an adaptation to training overload [17,18,19].

For this reason, despite the aforementioned information about ROSs in response to physical activity, there is no consensus of what the best post-workout recovery would be to minimize oxidative stress in the athletes’ body [6,9,10], especially because the physical exercises models and their evaluation methods have not been standardized, which makes a conclusive analysis difficult [6,9,10]. In this sense, the present study raised the hypothesis that using IBU during the recovery period of Paralympic Powerlifting athletes is beneficial for the parameters of sports performance, immunity maintenance and the reduction of oxidative stress.

In this regard, the objective of the present study was to analyze the effects of the use of ibuprofen on performance parameters, cell count and oxidative stress in national level Paralympic Powerlifting athletes in the period of resisted post-training recovery.

## 2. Materials and Methods

### 2.1. Study Design

The study design is shown in Figure 1. The study was carried out in three weeks, using the adapted bench press [20], with the first week being aimed at familiarization and the second and the third at the recovery method with the use of placebo (PLA) and Ibuprofen (IBU), the collections of the Peak Torque (PT), Rate of Torque Development (RTD), Fatigue Index (FI), Oxidative Stress Assessment through Thiobarbituric Acid Reactive Substances (TBARS) and Sulfidril Groups (SH), in addition to the blood indicators performed through the blood count and ammonia after training.

The order of the PLA or IBU conditions was determined randomly through a draw, considering 50% for each condition.

Week 1: familiarization; Week 2 and 3: recovery with the use of Ibuprofen or Placebo (week 2, 50% PLA and 50% IBU, changing in week 3). Pre-ingestion: 400 mg Ibuprofen/Placebo ingestion 15 min before training; Intervention: training length (3 h); Post-ingestion: 400 mg Ibuprofen/Placebo ingestion 5 h after training (5 h); Data collection: measures of strength in the Adapted Bench Press (FI: Fatigue Index, RTD: Rate of Torque Development, PT: Peak Torque), Oxidative Stress, TBARS: Thiobarbituric Acid Reactive Substances; SH: Sulfhydryl groups. After training, Ammonia and CBC were evaluated. 1-RM: One Maximum Repetition.

Collections were carried out between 9 a.m. and 12 p.m., according to the participants’ availability. All assessments were carried out 30 min before the training started and immediately after, 24 h and 48 h after the training. The participants were evaluated for the rate of torque development, peak torque and fatigue index. The blood variables evaluated were the cell count of the immune system and markers of oxidative stress {thiobarbituric acid (TBARS) and sulfhydryl groups (SH)} were performed before, after, two hours later, 24 h and 48 h after.

Before the intervention began, the athletes performed a previous warm-up for the upper limbs, using three exercises: (1) pulley elbow extension, (2) shoulders rotations with dumbbells, (3) shoulders abduction with dumbbells. Three sets of 10 to 20 maximum repetition (1-RM) were carried out; the warm-up lasted approximately 10 min [3,21]. Then, a specific warm-up was performed on the bench press itself with 30% of 1-RM where: 10 slow repetitions (eccentric 3-s × concentric 1-s) and 10 rapid repetitions (eccentric 1-s × concentric 1-s) were performed before the intervention started. It is noteworthy that during the specific warm-up, athletes received verbal encouragement to give their maximum performance [3,21].

Subsequently, the athletes were submitted to an intervention of five sets of five maximum repetitions (5 repetitions with 80–90% of 1-RM). In the intervention, the traditional method was applied, using only fixed loads (invariable resistance). Two types of recovery were applied: one using the wheat flour placebo and the other using the IBU (400 mg) where both groups ingested the tablet 15 min before and 5 h after training.

### 2.2. Sample

The sample was entirely composed of male athletes [20]. Forty percent of the athletes had spinal cord injury below the eighth thoracic vertebra, 20% had sequelae due to polio, 20% had a malformation of the lower limbs and 20% had disabilities due to brain injury. The athletes were of Brazilian nationality and competed on a national level with rankings in the top 10 of their respective categories. Exclusion criteria were adopted: (1) not participating in any phase of monitoring and data collection, (2) in the 24 h prior to the collection, strenuous exercise, (3) consumption of alcohol, caffeine, non-steroidal anti-inflammatory drugs (including IBU), nutritional supplements (confirmed by interview), (4) be allergic to Ibuprofen, (5) having any muscle or joint injuries and/or reporting a change in arterial hypertension.

The sample size was determined a priori based on a previous study [1], which found an effect size of partial squared eta (ɳ^2^*p*) = 0.6 for the analyses of the influence of ibuprofen on neuromuscular aspects in Paralympic Powerlifting athletes (in this case the variable was creatine kinase). Thus, the open-source G* Power software (Version 3.0; Berlin, Germany) was used in the statistical configuration for family tests “F” (ANOVA two way), considering an α < 0.05 and a β = 0.80. In addition, two groups (placebo x ibuprofen) in four distinct measures (Before × After × After 24 Hs × After 48 Hs) were considered. Thus, a minimum sample size of six subjects was indicated for the present study, with the sample power estimated at 0.80.

Table 1 shows the sample characterization.

### 2.3. Ethics

The athletes participated in the study voluntarily and signed a free and informed consent term, in accordance with resolution 466/2012 of the National Research Ethics Commission—CONEP, of the National Health Council, following the ethical principles expressed in the Helsinki Declaration (1964, reformulated in 1975, 1983, 1989, 1996, 2000, 2008 and 2013), by the World Medical Association. In addition, the present clinical trial was previously registered (CAEE ID: 79909917.0.0000.55.46) and approved by the Human Research Ethics Committee of the Federal University of Sergipe (UFS), under Statement Number 2637882/2018.

### 2.4. Body Mass Analysis

Body mass was measured while sitting on a Micheletti Electronic Wheelchair Scale (Model Mic Wheelchair) of the digital electronic platform type (Micheletti^®^, São Paulo, Brazil) with a maximum weight capacity of 500 kg (dimensions of 5.0 cm thickness, with a diameter of 102 × 120 cm).

### 2.5. Maximum Training Load Analysis

In order to determine the maximum training load, the 1-Repetition Maximum (1-RM) test was performed and, because the individuals evaluated were familiarized with the 1-RM test, they were not submitted to familiarization sessions. In the test, each subject started the attempts with a weight that could be lifted using maximum effort. Afterwards, weight increments were added until reaching the maximum load that could be lifted only once. If the practitioner was unable to perform a single repetition, 2.4 to 2.5% of the load used in the test were subtracted. The subjects rested 3–5 min between attempts [3,22].

### 2.6. Upper Limbs Muscle Strength

To measure muscle strength, the Fatigue Index (FI), the Peak Torque (PT), and the Rate of Torque Development (RTD) were determined by a Chronojump load cell (Chronojump^®^, BoscoSystem, Madrid, Spain), fixed on the Straight Bench Press, using Spider HMS Simond carabiners (Chamonix, France), with a breaking load of 21 KN, approved for climbing by the Union International des Associations d’Alpinisme (UIAA). A steel chain with a breaking load of 2300 kg was used to secure the load cell to the bench. The perpendicular distance between the load cell and the center of the joint was determined and used to calculate joint torques and fatigue index [21,23].

The isometric peak torque (PT) was measured by the maximum torque generated by the muscles of the upper limbs. PT was determined by the product of the peak isometric force, measured between the load cell cable fixation point and the adapted bench press, which was adjusted so that there was an angle close to 90° at the elbow, at a 15 cm distance from the starting point (chest to bar), verified with a device for measuring the angular amplitude, Model FL6010 (Sanny^®^, São Bernardo do Campo, Brazil). Participants were instructed to perform a single maximum movement until elbow extension (as fast as possible) and then relax, for PT evaluation.

As for the Fatigue Index (FI) assessment, the same exercise was performed and it was determined that the subjects maintained the maximum contraction for 5.0 s, where the index was determined by dividing the initial PT in relation to the final PT, subtracted from one. FI = {(Maximum PT − Minimum PT/Maximum Pt) × 100}. Thus, the results in Newton (Nm) were conceived by the formula Nm = (M) × (C) × (H), where M = Body mass in kg, C = 9.80665, H = Height of the bar in relation to the cell load (0.45 m), corresponding to the height that the equipment was fixed, adopting an angle of 90° between the forearm and the arm. The Rate of Torque Development (RTD) was determined using the Peak Torque to time ratio until reaching the Peak Torque (RTD = ΔPeak Torque/ΔTime), in 300 ms [21].

### 2.7. Blood Sample Collection

Blood samples were collected in the antecubital vein of the forearm and immediately transferred to tubes with EDTA. Blood collection (10 mL) was performed by a health professional (two nursing technicians). The samples were placed in vacuum blood collection tubes and sent to the Clinical Laboratory of the University Hospital of the Federal University of Sergipe (Aracaju, Sergipe, Brazil), where biochemical analyzes were performed by a laboratory biochemist.

### 2.8. Blood Cell and Leukocyte Count (HEMOGRAM)

Cell Dyn Ruby Abbott—The Cell-Dyn Ruby is an automatic hematology analyzer and multiparameter. The machine uses three basic methods. (1) The optical method is used to obtain red blood cell, platelet and leukocyte global counts. A photosensitive detector measures light scattering. The detected pulse length is proportional to the particle size (leukocyte, erythrocyte or platelet), which enables the identification of the volume of each of the formed elements of the blood. (2) In Laser Flow Cytometry, a flow of particulate matter passes through the laser beam crossing at an angle of 90 degrees, dispersing the light to a photomultiplier, which generates pulses in the histograms by determining the size and granularity of cells. This analysis is used to perform global and differential counts of leukocytes in 5 parts (neutrophils, lymphocytes, monocytes, eosinophils and basophils). (3) Colorimetry is a method that utilizes a chemical reaction color change and the final absorbance reading reaction, and is used for hemoglobin. Full blood counts were performed using a five-part differential hematology analyzer (Beckman Coulter AcT 5 diff AL Hematology Analyzer, CA, USA). The hematology analyzer uses a sequential dilution system and a dual-focused flow fluid dynamics technology, employing the Coulter Principle of impedance to count and measure the size of the cells.

### 2.9. Oxidative Stress

For oxidative stress, the tubes were centrifuged (3000× *g* for 10 min), and the plasma and serum were then aliquoted and stored at 4 °C for further analysis. To prevent the loss of volatile compounds, plasma ammonia was immediately measured using a spectrophotometric assay (Randox^®^, Crumlin, UK). The blood was centrifuged at 800× *g* for 15 min at 4 °C and the was serum stored at −80 °C.

In the serum, oxidative stress markers were evaluated. Lipoperoxidation was determined by measuring substances reactive to thiobarbituric acid (TBARS), according to the method described by Lapenna et al. [24]. For TBARS, 200 μL aliquots of the blood samples were added to a 400 μL mixture formed by equal parts of 15% trichloroacetic acid (TCA), 0.25N HCl and 0.375% TBA, plus 2.5 mM hydroxytoluene butylate (BHT) and 40 μL of 8.1% sodium dodecyl sulfate (SDS), being heated for 30 min at 95 °C in an oven. The pH of the mixture was adjusted to 0.9 with concentrated HCl. BHT was used to prevent lipid peroxidation during heating. After cooling to room temperature and adding 4 mL of butanol, the material was centrifuged at 800× *g* for 15 min at ±4 °C and the absorbance of the supernatant was measured at 532 nm. The molar extinction coefficient used was 1.54 × 105 M^−1^ cm^−1^ and the TBARS result was expressed in nmol Eq MDA/mL for the plasma samples.

The determination of sulfhydryl groups (SH) was carried out according to the methodology described by Faure and Lafond [25]. A 50 µL aliquot of the plasma was mixed in 1 mL of Tris-EDTA buffer (1 mM), and the first reading was taken at 412° (A1 reading). After this reading, 20 µL of 10 mM 5,5′-dithiobis 2-nitrobenzoic acid (DTNB) diluted in methanol were added. A new reading was taken after 15 min at room temperature (A2 reading). The Blank (B) contained only DTNB and Tris-EDTA buffer. The final unit was expressed in mM. The total sulfhydryl groups were calculated according to the molar absorption coefficient = 13,600 cm^−1^ M^−1^: (A2 − A1 − B) × 1.57 mM [25].

### 2.10. Post-Workout Recovery Using a Placebo

The control group received two sugar capsules, with packages identical to the IBU, 15 min before and 5 h after the resistance training. The same protocol as the IBU was followed, as described below.

### 2.11. Post-Workout Recovery Using Ibuprofen

This study followed the protocol used by De Souza et al. [16] and Fraga et al. [1], which consisted of administering IBU 15 min before and 5 h after resistance training. The experimental group received two capsules of IBU (400 mg) adding up to a total of 800 mg. Both IBU and PLA were packaged in identical capsules and the experiment was double-blind (i.e., participants and evaluators were unaware of what the capsules’ substance was). Upon receiving the capsules, all volunteers were instructed on the ingestion procedures. Follow-up calls from the research team ensured compliance.

### 2.12. Statistical Analysis

The normality of the data was tested by the Shapiro Wilk test and the assumption was not denied. Descriptive statistics were used with measures of central tendency, mean (X) ± Standard Deviation (SD). Comparisons with ammonia were performed using the paired Student’s t test. For the t-test, an effect size (Cohen’s “*d*”) was considered, adopting values of low effect (≤0.20), medium effect (0.20 to 0.80), high effect (0.80 to 1.20) and very high effect (>1.20) [26,27,28]. For performance comparisons between time periods (Before × After × After 24 Hs × After 48 Hs), the assumptions were complicit for the use of the ANOVA test (Two Way). Point differences were verified by Bonferroni’s Post Hoc. For ANOVA, the effect size was verified by the “partial squared eta” (ɳ2*p*), adopting values of low effect (≤0.05), medium effect (0.05 to 0.25), high effect (0.25 at 0.50) and very high effect (>0.50) [28]. All statistical treatment was performed using the computerized package Statistical Package for the Social Science (SPSS; version 22.0) considering that the level of significance adopted was *p* < 0.05.

## 3. Results

It is noteworthy that, based on the effect size results of the present study, the calculation of the sampling power through the open-source software G* Power software (Version 3.0; Berlin, Germany) was performed, considering an α < 0.05 and a β = 0.80. In this sense, the sample showed a power of >0.80 for the variables PT, FI, lymphocyte count, TBARS and SH.

Figure 2 presents the data related to the isometric strength through the Peak Torque (PT), Rate of Torque Development (RTD) and Fatigue Index (FI).

Regarding the results, the data presented point to: Figure 2A PT—“*” Indicates a difference (IntraClass) in Ibuprofen (IBU) between the moments 24 and 48 h later (*p* = 0.040, ɳ2*p* = 0.399, high effect), in PLA there were no differences. Figure 2B) RTD—“*” Indicates a difference (IntraClass) in Placebo (PLA) between the moments before and after (*p* = 0.003, ɳ2*p* = 0.542, very high effect). Figure 2C FI—“#” Indicates the difference between PLA and IBU (InterClass), at the moment 24 h later, (*p* = 0.015, ɳ2*p* = 0.745, very high effect). Regarding the supplement (IntraClass), in PLA—“a” Indicates a difference in the moments before and after; “b” in the moments before and 24 h after; “c” in the moments after and after 48 h and “d” in the moments 24 and 48 h afterwards (*p* <0.001). In the IBU supplement—“e” Indicates a difference between before and after; “f” in the moments before and 24 h after, “g” in the moments after and after 48 h and in the moments 24 and 48 h later (*p* < 0.001) and “h” in the moments after and 24 h after (*p* = 0.012). For PLA and IBU, ɳ2*p* = 0.982 (very high effect).

Table 2 shows the results of the blood count and blood variables, concerning to the changes in the cell counts of the volunteers’ immune system when comparing the post-PLA and the post-IBU.

The levels of C-Reactive Protein (CRP) increased (1.80 ± 1.47 to 3.55 ± 2.37 mg/dL, *p* = 0.031). There was no significant decrease in the total leukocyte count from 7.41 ± 1.80 to 6.64 ± 1.67 (mm^3^) (*p* = 0.415) and a raise in the percentage of neutrophils 3.72 ± 1.22 (%) for 4.88 ± 1.14 (%) (*p* = 0.151) did not suffer a statistical difference, the percentage of lymphocytes from 2.43 ± 0.58 to 3.48 ± 0.78 (%) was increased (*p* = 0.001). All values remained within the reference values for cell counts for the adult population.

Figure 3 shows Oxidative Stress (TBARS and SH) at different times with the use of a placebo (PLA) and Ibuprofen (IBU) at different times.

Regarding Oxidative Stress, the following differences were presented: Figure 3A TBARS, “#” Difference between PLA and IBU after 48 h (*p* = 0.010), “a” Difference in PLA between Before and 24 h after (*p* = 0.023), “B” Difference in PLA between 2 and 24 h after (*p* < 0.001), and “c” Difference in PLA between 24 and 48 h after (*p* = 0.034), ɳ2*p* = 0.173 (InterClass, medium effect) and ɳ2p = 0.479 (Intra Group, high effect). Figure 3B SH, “a” Difference in PLA Before and 24 h after (*p* = 0.030), and “b” Difference in IBU Before and 2 h after (*p* = 0.001), ɳ2*p* = 0.484 (IntraClass, high effect).

## 4. Discussion

This study aimed to analyze the effect of IBU on resisted post-workout recovery in PP athletes, by biomechanical variables and through biochemical indicators for muscle damage in the blood. The results highlighted that the Peak Torque with the use of IBU between 24 e 48 h after presented a significant difference, which resulted in better athlete performance. When evaluating the RTD, there was a decrease in the rate before and after training in the recovery method with PLA, and there were no differences in the IBU. The Fatigue Index was higher in recovery with the use of PLA after training compared to the use of IBU afterwards.

The results after the use of the IBU contributed to an improvement in the maximum isometric strength in relation to the use of the IBU 48 h after the training and the PLA 24 h after. A significant difference was also found with the use of the IBU 48 h after and PLA after the training. Therefore, it can be noticed that there was a maintenance of muscle function in the recovery with the use of IBU in the adapted bench press in Paralympic Powerlifting athletes concerning the PLA. This result can also be seen in the study by De Souza et al. [16] who demonstrated to mitigate fatigue in the gastrocnemius muscle in competing male runners who used IBU. Thus, when evaluating the FIM, the participants found better performance in the squat jump after the race than the control group.

The fatigue index showed significant differences in the results at all times of recovery. The moment that showed the highest peak was right after training with the use of PLA in relation to the use of IBU and in 48 h the values started to normalize. This result demonstrates that the recovery with the use of the IBU decreased the Fatigue Index and at the same time increased the Maximum Isometric Strength.

The muscle’s ability to generate strength can be decreased due to muscle fatigue, and damage the movement’s motor control [29,30]. It is noteworthy that the cause of fatigue can be central (i.e., when it affects the nervous system linked to muscle contraction) or peripheral (i.e., inhibitions in the contraction mechanisms of skeletal muscles). Therefore, fatigue when installed can disorder the movement sequence of the muscle segment’s movements [31], and the recovery becomes an important factor to observe [32].

In addition, this exercise protocol also demonstrated changes in the number of lymphocytes (immunological parameter). These data indicate that these extra blood cells were mobilized from the cell matrix because there was not enough time to produce new cells in the bone marrow [6]. The specific mechanisms by which leukocyte counts increase have been intensively discussed and some studies have suggested that exercise induces an increase in circulating stress hormones (growth hormone, epinephrine and norepinephrine) and that these hormones may play a role in the mobilization of white blood cells [33,34]. Increased production and the release of these hormones at the beginning of exercise can also stimulate the initial increase in the number of circulating leukocytes [35].

Recently, in a systematic review, Gonçalves et al. [6], exposed that many studies showed that intense physical activity increases the ROSs production in the human body. The results of this study do not show that intense physical activity, (represented here by the bench press) with five sets of five maximum repetitions (80–90% of 1-RM) was not capable to increase the ROSs production. In the present work, ROSs production was evaluated by the levels of TBARS and Sulfhydrys Group (SH). It is noteworthy that no other work had investigated the ROSs production using a similar protocol. Most studies use indirect methods to evaluate an increased ROSs production, for example, by measuring malonaldehyde (MDA), which is a marker of lipid peroxidation and reacts with thiobarbituric acid reactive substances (TBARS), signaling the existence of oxidative stress [36,37].

Barili et al. [38], found that the test on the treadmill was a sufficient stimulus to increase the peroxides production in elderly subjects. Wang et al. [39] investigated how the exercise intensity impacts redox status mediated by oxidation of Low-Density Lipoprotein (LDL) in monocytes. The aforementioned authors concluded the work by stating that high-intensity physical activity (80% VO_2_ max) increases ROSs production. Miyazaki et al. [40] investigated whether the high-intensity training (80% HRmax), during twelve weeks, would alter the oxidative stress induced by exercise after an event until the fatigue, verifying that exercising until the fatigue increases the ability of the neutrophils to produce ROSs and the training decreases this ability.

Studies measuring oxidative stress between different exercise models, such as aerobic exercise to fatigue and isometric exercise, and even associations between systemic oxidative stress, exercise intolerance and skeletal muscle abnormalities in patients with cardiac problems [41]. Another study comparing before and after with three different exercise protocols with trained subjects showed an increase of oxidative stress after intervention compared to pre-exercise [42]. Conversely, physical inactivity can reduce the body’s antioxidant systemic defense capacity [43].

It has also been shown that the immobilization of a leg for two weeks tends to induce the production of ROSs and impaired mitochondrial breathing capacity in the immobilized muscles [44]. Studies in humans indicate that exercise tends to be beneficial in the defense and prevention of oxidative stress, dependent on an inflammatory process [45,46] since, during exercise, the inner membrane of the mitochondria interferes with ROSs, and the intensity or volume of exercise leads to an impact in the activity of free radical production that can interfere with the degrees of oxidative damage [47]. It seems that only a single session of acute exercise is able to increase the total antioxidant capacity [42]. Muscle damage tends to induce the build-up of neutrophils and cytokines, inducing oxidative stress [46]. On the other hand, researches indicate that chronic physical activities tend to increase adaptive and antioxidant defense systems [47,48].

Regarding the increase in free radicals, there is an indication that the antioxidant activity in the body tends not to decrease after intense chronic and acute exercises [46]. De Souza et al., [49] demonstrated lipid peroxidation in high intensity and long duration exercises in healthy individuals. Plasma MDA levels were measured before and after exercise until fatigue and did not undergo any significant changes. In the same direction, high intensity or exhaustive strength exercises tend to cause injuries and chronic fatigue. This would happen due to the imbalance between the production of reactive oxygen species (ROSs) and the endogenous antioxidant activity. Although ideal ROS production is important for muscle contraction, high ROSs concentrations tend to promote exercise-induced fatigue [50,51].

Skeletal musculature is reported to produce greater amounts of superoxide anion during training [52]. However, improvements have been reported concerning the oxidative stress provided by strength training [53,54]. Nevertheless, a higher training volume tends not to alter oxidative stress markers [55]. In this direction, studies indicate that in trained weightlifting athletes, high-intensity strength training tends to increase oxidative stress and decrease the antioxidant capacity of these athletes [56], which tends to lead to unfavorable effects of exercise in relation to health. In training with loads above 70% of 1 RM, the oxidative stress markers did not change. In contrast, high-intensity strength training, such as the one in the study, tends to increase the level of oxidative markers, as well as tends to decrease the production of antioxidants in powerlifting athletes [56], despite moderate to high-intensity training tends to improve oxidative stress [53,54]. Thus, it appears that strength training tends to improve oxidative stress among athletes [57]. The use of anti-inflammatory drugs, such as ibuprofen, tends to delay the anti-inflammatory response after exercise, helping the performance of powerlifting athletes [1], and this would explain the decrease in fatigue in the condition with ibuprofen use found in our study.

As is already widely discussed in the literature, high intensity or exhaustive physical exercise is recognized for increasing oxygen consumption resulting in a greater formation of reactive oxygen species (ROSs), greater susceptibility to muscle injuries and chronic fatigue [58]. In turn, non-steroidal anti-inflammatory agents (NSAIDs) became the most widely prescribed and used drugs worldwide [59,60], the use of IBU Non-steroidal anti-inflammatory drugs (NSAIDs) constitute one of the most consumed drug classes in the world. They have analgesic, antipyretic and anti-inflammatory effects that are used to treat acute pain arising from inflammation. Its effects occur through the reduction of the enzyme cyclooxygenase (COX), resulting in a decrease in precursors of prostaglandins and thromboxanes. The use of NSAIDs, when administered orally, is generally rapidly absorbed, it was found that the 400 mg tablet of IBU showed a peak concentration of 20–40 mg/mL in 1–2 h and decreasing to 5 mg/mL at the end of 6 h [61].

In this sense, the rapid absorption of IBU, which leads to rapid lowering of (MDA or TBARS) levels, occurs because it is subject to N-hydroxylation in the liver with the involvement of cytochrome P450 enzymes to form a toxic metabolite (NAPQI), which is rapidly inactivated by glutathione sulfhydryl (GSH) groups [62]. In large amounts of NAPQI, there is depletion of endogenous GSH in the liver and favors the binding of NAPQI with cellular biological macromolecules, such as proteins, nucleic acids and lipids, resulting in mitochondrial damage, endoplasmic reticulum stress and necrotic cell death. Then, in the toxicity phase, mitochondrial dysfunction increased oxidative stress occurs (damaged mitochondria lead to overproduction of reactive oxygen species (ROSs) [63,64]. As previously mentioned, the prophylactic use of IBU has a rapid absorption by the body, and as the levels of (MDA or TBARS) remain high as shown 48 h later. Finally, in studies carried out with animals that used ibuprofen, a cyclooxygenase inhibitor, the hematocrit and platelet counts were similar to those that did not receive ibuprofen [65].

As previously shown, the results of the present study agree with several other studies. However, it is necessary to emphasize that further research is still needed to recognize which model of physical activity really increases the production of ROSs without being neutralized by the antioxidant defense system, resulting in oxidative stress, which could cause damage to biomolecular structures.

Nevertheless, despite the important findings, the present study has limitations. The absence of monitoring food and quality of sleep of athletes during the recovery period made it impossible to control the intake of any food that would help to reduce oxidative stress and in the approach regarding sleep interference on oxidative stress.

## 5. Conclusions

It was concluded that the recovery with the use of Ibuprofen (IBU) presented a lower fatigue index and a lower decrease in strength when compared to the recovery with the placebo (PLA). There was also a reduction of muscle damage with the use of IBU compared to the recovery with PLA. This study demonstrated that, in the two forms of recovery, with PLA and with the use of the IBU after strength training, there was no protective effect of the anti-inflammatory on oxidative stress markers.

From the point of view of practical applications of the findings, the results indicate that the use of ibuprofen can be a good strategy in the recovery of athletes aiming for a new training session, when it is necessary to recover more effectively after a workout, and even for competitions, as it allows an improved recovery in relation to the recovery without the use of supplements.

## Figures and Tables

**Figure 1 biology-10-00986-f001:**
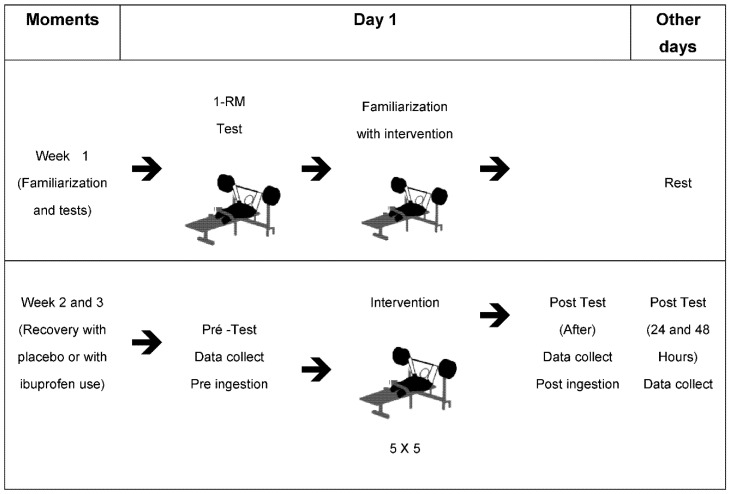
Experimental drawing. Weekly training schedule.

**Figure 2 biology-10-00986-f002:**
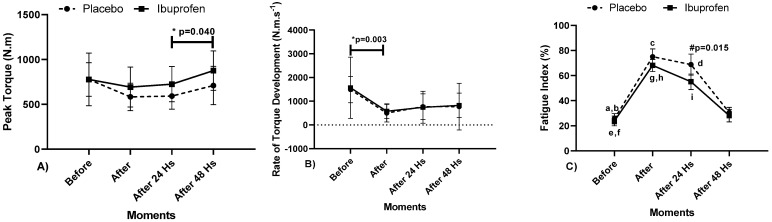
(**A**) Peak of Torque (PT), (**B**) Rate of Torque Development (RTD) e (**C**) Fatigue Index (FI) in diverse moments with Placebo (PLA) and Ibuprofen (IBU) used in recovery. Legend: “*”: Indicates IntraClass difference in (**A**,**B**); “a–h”: Indicates IntraClass differences in (**C**) and “#”: Indicates InterClass difference (**C**) (*p* < 0.05).

**Figure 3 biology-10-00986-f003:**
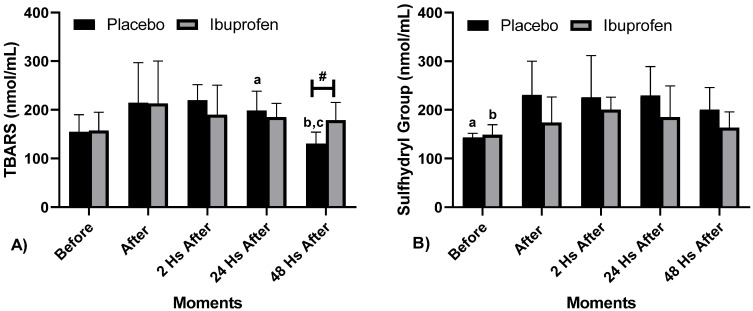
Oxidative Stress (**A**) Thiobarbituric Acid Reactive Substance (TBARS) e (**B**) Sulfhydrys Group (SH), at diverse moments with Placebo (PLA) and Ibuprofen (IBU) use at recovery. Legend: “a–c”: Indicates IntraClass differences, and “#”: Indicates InterClass difference C) (*p* < 0.05).

**Table 1 biology-10-00986-t001:** Sample characterization.

Variables	(Mean ± Standard Deviation)
*n*	10
Age (years)	27.13 ± 5.57
Body Weight (kg)	79.25 ± 25.51
Experience (years)	2.99 ± 0.51
1-RM/Bench press (kg)	137.13 ± 30.53 *
1-RM/Body Weight	1.80 ± 0.31 **

* All athletes with loads that keep them in the top 10 of their categories nationwide. ** Athletes with values above 1.4 in the Bench Press (1-RM/Body Weight) would be considered elite athletes, according to Ball & Wedman, [22].

**Table 2 biology-10-00986-t002:** Blood markers in the presence and the absence of Ibuprofen.

Variables	Placebo	Ibuprofen	*p*	Cohen’s d
CPR (mg/dL)	1.80 ±1.47	3.55 ± 2.37	0.031 *	1.80 d
Urea (mg/dL)	28.88 ± 6.66	24.38 ± 6.74	0.074	3.35 d
Uric acid (mg/dL)	5.14 ± 1.10	5.50 ± 1.21	0.140	1.41 d
Leukocytes (mm^3^)	7.41 ± 1.80	6.64 ± 1.67	0.415	2.08 d
Neutrophils (%)	3.72 ± 1.22	4.88 ± 1.14	0.151	4.66 d
Lymphocytes (%)	2.43 ± 0.58	3.48 ± 0.78	0.001 *	4.36 d
Erythrocytes (million/mm^3^)	5.06 ± 0.39	5.13 ± 0.46	0.221	0.64 b
Hemoglobin (g/mL)	15.08 ± 1.12	15.00 ± 1.43	0.767	0.20 a
Hematocrit (%)	42.63 ± 3.30	43.95 ± 4.00	0.019 *	1.31 d
MCV (U3)	84.29 ± 2.21	85.67 ± 3.36	0.090	1.08 c
MCH (UUG)	29.79 ± 0.52	29.23 ± 0.89	0.123	1.42 d
MCHC (%)	35.36 ± 0.94	34.12 ± 1.14	0.007 *	4.31 d
RDW (%)	10.24 ± 3.35	11.50 ± 0.54	0.305	0.45 b

* *p* ≤ 0.05 (ANOVA two way and Post Hoc de Bonferroni). “a” small effect (≤0.20), “b” medium effect (0.20 a 0.80), “c” high effect (0.80 a 1.20) and “d” very high effect (>1.20). Legend: MCV: Mean Corpuscular Volume, MCHC: Mean Corpuscular Hemoglobin Concentration, RDW: Erythrocyte anisocytosis index.

## Data Availability

The data that support this study can be obtained from the address: https://www.ufs.br/Department%20of%20Physical%20Education, accessed on 12 July 2021.

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
