# Peer review of "Effects of Ibuprofen Use on Lymphocyte Count and Oxidative Stress in Elite Paralympic Powerlifting"

_biology, 2021, doi:10.3390/biology10100986_

Round 1

Reviewer 1 Report

This study investigated the effects of the use 800 mg of ibuprofen and placebo on performance parameters, cell count and oxidative stress in 10 paralympic powerlifting athletes in the period of resisted post-training recovery. The authors described that ibuprofen had a positive effect on strength 45 indicators, with decreased fatigue and increased lymphocyte count. However, ibuprofen was not able to protect from oxidative stress.  Although the search for identify medication to improve athletic parameters are valuable in this field, I have some concerns that prevent me to endorse its acceptance at the present stage. The major of them are the results and discussion of the results, which should be expanded to explain the results obtained. In addition, English should be improved. Below are more specific comments.

Pag2, line 96-99 the authors reported that “In this sense, the present study suggests that….reduction of oxidative stress”. However, the authors first suggest that the results did not affect the oxidative stress (Page 1, line 47-48). What is correct?

Page 7, lines289-295, Results description is confuse. This should be rewritten.

Page 8, line : “There was an increase in the total leukocyte count from 7.41 ± 1.80 to 6.64 ± 1.67 308 (mm3) (p = 0.415)”… This is not correct. There is a decrease in leukocyte count. However,  this is not statistical significant.

Pag9, line 328 : The authors reported that “ This study aimed to analyze the effect of IBU on resisted post-workout recovery in  Paralympic Powerlifting athletes, by biomechanical variables, imaging exams, and through biochemical indicators for muscle damage in the blood.”  However, the authors did not show imaging exams.

Page 9, line 332 : “When evaluating the TDF, a decrease  in the rate before and after training “. Again, the authors did not show the results cited.

Page, line 333: “was found “  why?

How authors explain that after 24h of ibuprofen administration the results show increased TBARS in placebo group and after 48 h of ibuprofen administration opposite results were observed?

Why the authors are using the sulfhydryl group as biomarker to oxidative stress?

Other biomarkers as carbonyl protein as well as oxidative enzyme activities (superoxide dismutase, catalase, glutathione peroxidase) may help to understand the involvement of ibuprofen in oxidative stress.

The discussion about oxidative stress is confusing and should be better explained.

There are some abbreviations such as TDF and FIM that were not described before.

Author Response

Thank you for your comments regarding the manuscript. Initially we apologize for the delay, but due to the English revision we ended up needing a little more time. We respond to all inquiries and suggestions from reviewers. To make the answers easier to understand, we've put each caption in a different color. For reviewer 1, we used the color yellow, for reviewer 2, the color blue, for reviewer 3, the green color and the purple color was used for English review. On this occasion we renew our protests of consideration and high esteem, respectfully

Reviewer 2 Report

The authors propose that ibuprofen pre and post powerlifting can be associated with better lymphocyte response and oxidative stress balance, but they do not find any relations with the initial purpose. 

The authors need to improve the manuscript, for a better understanding in:

Abstract
  Define "RM" and inform the IBU concentration

They use IF (line 55) and FI (line 53), please standardize it

Introduction
 Line 70. Please change the word “sh..” by waste (formal forms)
Line 83-84 link with a full stop  (period or dot followed) instead at full stop, new paragraph

Methods
Line 107-108: please explain the concepts
Line 116: acquaintance???  I do not understand the idea of using that word.

Line 117: all the athletes receive Ibu pre and post? How many were placebo, did one of them only receive one doses?

Line 242: why you use the word “organs”, all the paragraph is weird.

What was the references for to choose the IBU doses and time administration?
Figure 1 it says “pós”???? please correct it
Table 1:  the gender of the participants is? The BMI?

How the authors control the fact that PLA receives sugar (a nutrients thar modified metabolism) and it could be an interferent?

Results
G*Power and sampling was not informed in methodologies
Line 293:  the is a capital H in the text
Line 286 -287 It is not understood what the authors wanted to inform

Why IBU increases hematocrit? Please discuss it
Please explain better Table 2

Discussion
Line 328: change “Paralympic powerlifting” by “PP”
Why you do not assay CK as a factor of muscle damage?
How the authors explain the significance increase of TBARs at 48h of IBU? It must be largely discussed.

Author Response

(The authors gave the same response as above.)

Reviewer 3 Report

The manuscript entitled “Effects of ibuprofen use on lymphocyte count and oxidative stress in elite paralympic powerlifting.” This work is merit for publication at Biology after some major modification. So I have some points that may help to improve the work as follows:

1-Abstract is good but need more explain about the main aim of work

2- The introduction should be extended to discuss the hypothesis and research questions in details. Additionally, the introduction should cover the recent literature related to this subject.

3- Material and methods

The methodologies should be explained in details so that the results are reproducible.

4-Results

The results are clear and important.

5-Discussion
The discussion section still needs improvement, and should be linked to the findings of the previous reports on this topic.

5- The conclusion

A section for conclusions need more explain and should include the most significant findings and future works only.

6- English writing should be checked by a native English speaking expert.

Author Response

(The authors gave the same response as above.)

Round 2

Reviewer 2 Report

The authors made all the requested corrections.

For me it`s ok.

Reviewer 3 Report

The authors have made changes to the manuscript, so I consider it can be accepted for publication.